# Mental health and quality of life following breast cancer diagnosis in patients seen at a tertiary care hospital in Nairobi, Kenya: A qualitative study

mental health; breast cancer; stigma; Kenya

**Corresponding author:**
Divya Annamalai;
Email: divyaa22@uab.edu

Divya Annamalai[1] (ID), Anna Helova[1,2], Mansoor Saleh[3,4], Nancy Gikaara[3], Sehrish Rupani[3], Innocent Abayo[3], Noureen Karimi[3], Karishma Sharma[3], Isaiah Omedo[3], Kevin Owuor[1,5], Lily Gutnik[6] and Janet M. Turan[1,2,7]

[1]Sparkman Center for Global Health, University of Alabama at Birmingham, Birmingham, AL, USA; [2]Department of Health Policy and Organization, School of Public Health, University of Alabama at Birmingham, Birmingham, AL, USA; [3]Clinical Research Unit, Aga Khan University Cancer Center, Aga Khan University Hospital, Nairobi, Kenya; [4]Department of Hematology and Oncology, Aga Khan University Hospital, Nairobi, Kenya; [5]Department of Biostatistics, School of Public Health, University of Alabama at Birmingham, Birmingham, AL, USA; [6]Department of Surgery, UAB Heersink School of Medicine, University of Alabama at Birmingham, Birmingham, AL, USA and [7]Department of Public Health, School of Medicine, Koç University, Istanbul, Turkey

## Abstract

Mental health challenges are common following cancer diagnosis, negatively impacting treatment and quality of life for breast cancer (BC) patients. This pilot study provides an understanding of the impacts of BC diagnosis and care experiences on the mental health of patients seen at the Aga Khan University Hospital in Nairobi, Kenya. We conducted 40 in-depth interviews, including 10 women with newly diagnosed BC, 10 women with metastatic BC, 10 family members and 10 healthcare professionals. Data were transcribed, translated into English as needed and coded using Dedoose software. Following BC diagnosis, it was reported that patients faced various physical, social, psychological and spiritual factors affecting their mental health and quality of life. Our interviews with each group indicated that BC patients experienced feelings of stress, anxiety and depression related to treatments and accompanying side effects. Disclosure concerns, financial impacts, relationship strain and negative outlooks on life were common among BC patients. The findings indicate that BC diagnosis and care experiences influence mental health in this population. With this basis, understanding and addressing the mental health challenges of BC patients is crucial to improve mental health and quality of life.

## Impact statement

Amid the limited research on the topic, our pilot study provides novel insight into the impact of breast cancer (BC) diagnosis and care experiences on the mental health of patients seen at the Aga Khan University Hospital in Nairobi, Kenya (AKUHN). Through conducting interviews with patients, their family members and healthcare professionals at AKUHN, we have identified various physical, social, psychological and spiritual factors that influence patient mental health. Our findings highlight that patients experience stress, anxiety and depression during and following cancer treatment as well as negative outlooks on life and thoughts about death. Patients expressed worry about leaving family behind, reaction to BC disclosure, financial impacts and spousal relationship strain. Many BC patients described feeling like a burden and distress, highlighting a need for increased cancer support services. We hope this study facilitates more research surrounding mental health among cancer patients in Kenya, Africa and in other low- or middle-income countries.

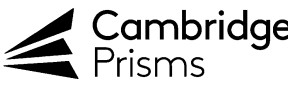



## Introduction

Breast cancer (BC) is the most diagnosed cancer worldwide and a growing public health burden in Sub-Saharan Africa (SSA) (Arnold et al., 2022). In 2020, there were an estimated 2.3 million new BC cases globally, with 129,400 of those cases in SSA (Bray and Parkin, 2022; Sung et al., 2021). While BC mortality is on the decline in many high-income countries, it appears to be on the rise in SSA (Anyigba et al., 2021; Rauniyar et al., 2023). This may be due to the greater focus on combating infectious diseases in SSA, leading to lower priority given to addressing cancer

(Ba et al., 2020). This could also be a result of inadequate access to quality cancer tertiary centers, limited screening programs and lack of knowledge among primary care clinicians in SSA (Pace and Shulman, 2016). Patients' limited knowledge of BC symptoms, fear of cancer diagnosis, financial concerns and use of traditional healers are also linked to late cancer diagnosis in SSA (Espina et al., 2017).

Though BC has a significant presence in many SSA countries, its impact on mental health remains understudied (Kagee, 2022). Irrespective of stage, having BC in any setting can pose new burdens and stressors that may impact patient mental health (İzci et al., 2016). The few existing studies suggest that BC diagnosis may be associated with psychosocial impacts in SSA countries. In one study in Ghana, researchers reported that BC diagnosis and treatment led to physical and emotional problems that interfered with patient quality of life (Iddrisu et al., 2020).

Various sociocultural factors in SSA can impact the BC patient care experience. The healthcare sector in many SSA countries is characterized by limited knowledge of BC screening methods, access to screening facilities and trained healthcare professionals (HCPs), contributing to delays in early cancer diagnosis (Akuoko et al., 2017; Omotoso et al., 2023; Tetteh and Faulkner, 2016). Limited awareness of BC in SSA also contributes to poor disease perceptions, with many associating it with death (Tetteh and Faulkner, 2016). With treatment often not being covered by National Health Insurance, patients also pay high out-of-pocket fees for treatment (Omotoso et al., 2023; Tetteh and Faulkner, 2016).

Physical impacts from cancer treatment can also greatly burden a patient, leading to negative effects on one's mental state and mood (İzci et al., 2016). For instance, young women in Ghana experienced severe body weakness and pain that incapacitated them or felt unattractive following mastectomy (Iddrisu et al., 2020). Fears of death or loss of their femininity from mastectomy and chemotherapy are also prevalent in SSA (Lambert et al., 2020). In one study focusing on African women in SSA, fear of mastectomy and death from BC prevented women from consulting their physicians or seeking treatment for their illness. In one study focusing on married BC patients in Nigeria, 70% of patients felt less feminine, had negative body image, or felt undesirable to their spouses after mastectomy (Tetteh and Faulkner, 2016).

In addition to these impacts, community stigma, fears of leaving family/children behind, employment burdens, related disclosure concerns and negative outlooks on life may worsen pre-existing mental health issues and increase the risk of psychological disorders after BC diagnosis (İzci et al., 2016).

In Kenya, BC exists as the most common type of cancer, with 7,243 new cases reported in 2022 (Ferlay et al., 2024). It is predicted that by 2025, the incidence and mortality of BC in Kenya will increase by 35% (Gakunga et al., 2019). Despite efforts to promote BC screening, studies indicate that only about 5% of Kenyan women undergo BC screening, which contrasts with the approximate 70% rate in the United States. BC private treatment costs also significantly exceed the average annual household expenditure per Kenyan adult (Subramanian et al., 2019; Subramanian et al., 2018).

Kenya has made significant strides towards cancer care over the years, despite having a relatively limited number of cancer care facilities. Public institutions such as Kenyatta National Hospital, the country's largest referral hospital, provide surgery, radiotherapy and chemotherapy services. Other public hospitals, such as Moi Teaching and Referral Hospital and Coast General Hospital, provide only surgery and chemotherapy. Private facilities, like Aga Khan University Hospital in Nairobi (AKUHN), provide all forms of cancer treatment (Makau-Barasa et al., 2018).

There are several government-supported cancer advocacy groups in Kenya. The National Cancer Control Program coordinates country-wide efforts for cancer prevention, early detection, treatment and palliative care (Manirakiza et al., 2023). Additional advocacy groups such as the Kenya Network of Cancer Organizations and the Kenya Hospices and Palliative Care Association advocate for improved cancer care policies and patient support initiatives (Ali, 2016; Morgan et al., 2018). However, despite the strides made towards cancer care in Kenya, BC continues to be an urgent and substantial issue.

There are gaps in our knowledge about how a BC diagnosis affects mental health and quality of life in African settings. In Kenya, it has been observed that increasing the severity of cancer can lead to a greater risk of psychological issues (Ndetei et al., 2018). This is significant as 30% of early-diagnosed BC patients eventually develop metastatic cancer and the specific needs and impacts on these patients have been neglected (Shaikh et al., 2022). With the heightened mental health risk of BC patients in Kenya, it is important to understand what factors contribute to mental health outcomes.

This paper reports on the qualitative findings of a larger mixed methods pilot study and aims to provide an understanding of the experiences, mechanisms and effects of BC diagnosis on the mental health and quality of life of women seen at AKUHN, a tertiary university hospital in Nairobi, Kenya. This qualitative analysis also aims to understand the sociocultural factors that influence BC patients' mental health within this Kenyan context based on the perspectives of BC patients, BC patient family members and HCPs. The relationship of BC patients to their family members is important as family members often assume the patient's caregiving role due to sociocultural obligations (Kusi et al., 2020). HCPs follow patients through various stages of their treatment and have additional perspectives on patient care experiences (Tshabalala et al., 2023). The findings will serve as preliminary data for intervention studies aimed at mitigating the mental health effects of BC diagnosis, promoting awareness and improving BC patient quality of life and health outcomes.

## Methods

### Setting

This study is a collaboration between the University of Alabama at Birmingham (UAB) and AKUHN, a private, 200-bed, multi-specialty tertiary care hospital in Nairobi, Kenya. AKUHN patients come from various socio-economic backgrounds. Services are paid for through a mixture of coverage by the National Health Insurance Fund (NHIF), employment-based insurance, self-pay and AKUHN's Welfare Program.

### Data collection procedures

Inclusion Criteria were (1) patients with BC [newly diagnosed (ND) within the past 6 months of diagnosis ($n = 10$) or patients diagnosed with metastatic BC (MS) on active treatment or in post-treatment follow-up ($n = 10$)]. (2) Family members of BC patients treated at AKUHN, male or female ($n = 10$). (3) HCPs caring for BC patients (physicians, nurses, pharmacists, management, or other healthcare professionals) at AKUHN ($n = 10$). The sample size was informed by estimated data saturation. This determination was

based on a prior study stating that saturation of in-depth interview data can be achieved in a range of interviews between 9 and 17 (Hennink and Kaiser, 2022).

Family members are defined as caregivers of BC patients accompanying the patient at clinic visits. HCPs were those involved in providing BC care and treatment to patients at AKUHN. All participants were 18 years of age or older, able to participate in an interview in English or Swahili, and able to provide informed consent.

Signed informed consent was obtained from all participants. BC patient and family interviews were conducted in person in English or Kiswahili by a male or female staff member at AKUHN. Interviews were conducted in a private room and audio was recorded using digital recorders. HCP interviews were conducted remotely through Zoom by a female member of the UAB study team. A Zoom link was sent to participants at least 48 hours before the interview, and participants gave verbal consent to audio recording. Each interview was 60–90 minutes. All interviewers were trained to conduct qualitative interviews and followed one of three in-depth interview guides designed for each participant group by the study team.

Interview topics focused on the impact of a BC diagnosis on the life of a patient with questions relating to health outcomes and stressors, quality of life, burdens and fears resulting from the diagnosis. Other topics included household information, disclosure, social support, experiences of stigma and the effects of BC on a patient's life.

### Data management and analysis

Participant characteristics were descriptively presented using counts and percentages or medians with interquartile ranges (IQR). Audio files were password-protected, de-identified, and uploaded to a computer. These audio files were transcribed onto Microsoft Word in the language of the interview and translated into English by an experienced bi-lingual (English, Kiswahili

transcriptionist. Coding and analysis were conducted by coders from the UAB and AKUHN research teams.

We utilized a reflexive thematic analysis approach to coding and analysis, as described by Braun and Clarke (Braun and Clarke, 2021). Dedoose software was utilized for coding and analysis (Huynh, 2021). A preliminary coding scheme was initially developed based on topics from existing literature, guides, and emerging findings from the transcripts. The scheme was refined, and transcripts were coded broadly, followed by second-level fine coding, using a combined inductive-deductive strategy in which themes from theory and literature were supplemented with emerging themes from collected data (Braun and Clarke, 2021). We utilized the Quality of Life Model (Figure 1) as a lens through which to interpret our data on the physical, social, psychological, and spiritual factors that may contribute to the mental health and quality of life of BC patients (Mollica and Newman, 2014).

This paper intentionally focuses on understanding the effects of BC diagnosis and treatment experiences on the mental health and quality of life of patients seen at AKUHN. We aimed to achieve this through interviews to obtain the diverse perspectives of BC patients, their family members and their healthcare providers.

### Results

We conducted 40 in-depth interviews (10 ND, 10 MS, 10 family members, 10 HCPs). The characteristics of the study participants are presented in Table 1. The majority of participants had more than secondary education, were married, were Christian and were of African ethnicity.

We organized our findings utilizing the Quality of Life Model (Figure 1). Main themes within the model domains included the following: overall physical health decline and pain (physical domain); family disruption, negative impact of relationship with a male partner, impact on appearance and self-esteem, financial burden (social domain); mental health impacts on patients with BC, negative outlook on life following BC diagnosis (psychological

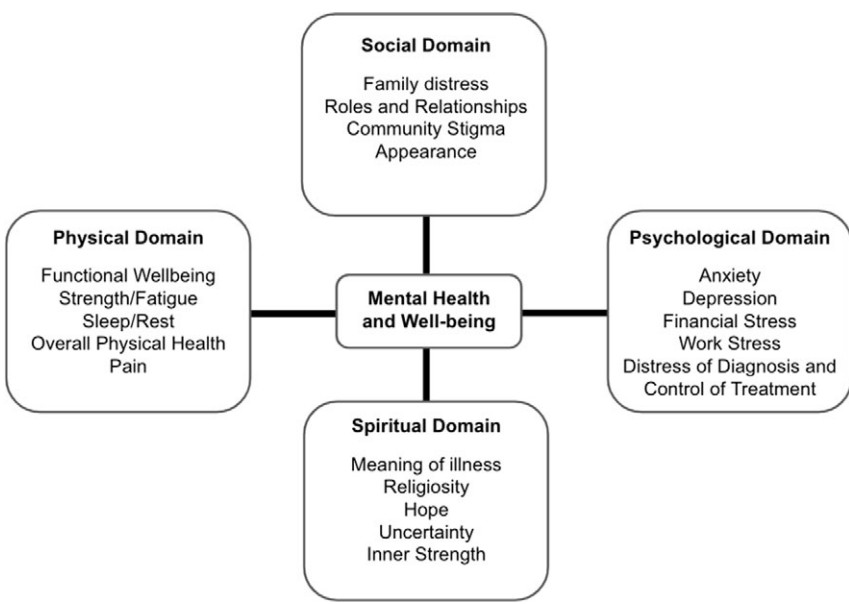

**Figure 1.** Quality of life model applied to BC patients in Nairobi, Kenya.

**Table 1.** Sociodemographic and BC related characteristics of interview groups. Represented using counts with percentages or medians with interquartile ranges (IQR)

| Characteristics | Patients (*n* = 10 Metastatic BC, *n* = 10, Newly diagnosed) | Newly diagnosed BC patients (*n* = 10) | Metastatic BC patients (*n* = 10) | Family members (*n* = 10) | Healthcare professionals (*n* = 10) |
|---|---|---|---|---|---|
| Age: Median (IQR) | 45 (37–50.5) | 43(37.8–52.5) | 45 (38.8–48.3) | 39 (31–44.8) | 32 (30–38.5) |
| Gender (%) | | | | | |
| Female | 20 (100.0%) | 10 (100.0%) | 10 (100.0%) | 5 (50.0%) | 5 (50.0%) |
| Male | 0 (0.0%) | 0 (0.0%) | 0 (0.0%) | 5 (50.0%) | 5 (50.0%) |
| Educational Level (%) | | | | | |
| Completed Secondary | 1 (5.0%) | 0 (0.0%) | 1 (10.0%) | 0 (0.0%) | 2 (20.0%) |
| More Than Secondary | 19 (95.0%) | 10 (100.0%) | 9 (90.0%) | 10 (100%) | 8 (80.0%) |
| Marital Status (%) | | | | | |
| Single, never married | 7 (35.0%) | 4 (40.0%) | 3 (30.0%) | 2 (20.0%) | 5 (50.0%) |
| Married | 12 (60.0%) | 5 (50.0%) | 7 (70.0%) | 7 (70.0%) | 4 (40.0%) |
| Widowed/divorced/separated | 1 (5.0%) | 1 (10.0%) | 0 (0.0%) | 1 (10.0%) | 0 (0.0%) |
| Religion (%) | | | | | |
| Christian | 13 (65.0%) | 8 (80.0%) | 5 (50.0%) | 8 (80.0%) | 9 (90.0%) |
| Islam | 5 (25.0%) | 1(10.0%) | 4 (40.0%) | 2 (20.0%) | 1 (10.0%) |
| Other | 2 (10.0%) | 1(10.0%) | 1(10.0%) | 0 (0.0%) | 0 (0.0%) |
| Number of Living Children: Median (IQR) | 2 (1–3) | 2 (1–3) | 2.5 (1.3–3) | 2 (1.3–2) | 1 (0–1.8) |
| Ethnicity | | | | | |
| African | 14 (70.0%) | 8 (80.0%) | 6 (60.0%) | 9 (90.0%) | 9 (90.0%) |
| Asian | 5 (25.0%) | 1 (10.0%) | 4 (40.0%) | 1 (10.0%) | 1 (10.0%) |
| Other | 1 (5.0%) | 1 (10.0%) | 0 (0.0%) | 0 (0.0%) | 0 (0.0%) |

domain); and relationship with religion during BC diagnosis and treatment (spiritual domain).

### Physical domain: "This pain is too much" (MS, 37, Female)

#### Treatment side-effects

Patients expressed that available treatments (chemotherapy, radiotherapy and surgery) and side effects (vomiting, fatigue and body pain) negatively affected their physical well-being. Patients and family members described these side effects as difficult and traumatizing consequences of treatment.

> "Chemo is a bit tough. Tough in the sense that it's like taking your whole life and replacing it with something so strange, so painful, and all sorts of complications...."
>
> (ND, 41, Female)

#### Functional well-being

We found that BC treatment and its side effects can lead to physical impacts that hinder a patient's ability to perform household duties and a job. The level of the impact depends on treatment, response to treatment and the severity of side effects. Many patients rely on familial assistance for these tasks.

> "She's always been someone who's on the go and doing things for her kids and family. And then suddenly that person is slowed down and can't get out of bed. You know? Can't even cook a meal."
>
> (Family of ND, 45, Female)

#### Pain

Pain and symptom management was a major factor described in a patient's care experience. Pain is experienced by both ND and MS patients but was more pronounced in interviews with MS patients.

> "There are some times you wake up, you're feeling very strong. And you can do almost everything. And then there are times when you wake up, you cannot do anything, literally you just want to sit and sleep and do nothing at all. …This journey is too long; this journey is too hard. This journey is just depressing."
>
> (MS, 45, Female)

### Social domain: "I don't think I can win this battle alone" (ND, 40, Female)

#### Social support

Many patients described receiving emotional support and assistance from their close family members and/or spouses.

> "My husband every day in the morning, makes sure my breakfast is ready, my juice is ready, and he's just there for me. He's so caring and supportive."
>
> (MS, 49, Female)

In some cases, the news of the BC diagnosis had a great emotional impact on close family members and the patient's spouse. Some family members expressed the feeling of not doing enough for the patient. HCPs discussed how the lengthy process of cancer care can burden the family's ability to provide support. HCPs also mentioned

the need for more training on how to provide emotional support to patients.

> "Sometimes I'm the one to feel burdened and I feel like I'm not doing enough. I do ask myself why when she's alone, she cries… maybe there's something. She might get something better from somebody else…but I'm doing my best. She knows I'm doing my best."
>
> *(Family of MS, 34, Female)*

> "I would still just need to know a bit more training on how to approach such a situation where a patient might actually be going into end-of-life care… it'd be very handy to have such a training."
>
> *(HCP, 30, Female)*

Several BC patients shared that their male partners went through a period of denial and delayed conversation on the topic. Some patients expressed that they understand if their male partners seek "comfort" elsewhere. HCPs recounted instances of spousal abandonment for BC patients and the importance of family support in improving BC patient outlook on life.

> "I've noticed a lot of family issues. Some women have been kind of left alone by their family, especially I guess, their husbands because they realize it's difficult to look after the patient while they're doing this. And we've seen quite a lot where if a patient is married, sometimes their husband will just leave. If they have kids, they now have to struggle with the single parenthood, as well as now this new diagnosis."
>
> *(HCP, 30, Female)*

### Disclosure concerns

Fear of negative reactions from close family members and fears surrounding disclosure in some cases caused patients to isolate themselves as not to be a burden to their family.

> "Who do I tell, I cannot tell my family members that I am thinking about death, they will not take it well."
>
> *(MS, 37, Female)*

Generally, patients tended to disclose their diagnosis only to family members or trusted friends. There was a general unwillingness to publicly talk about one's cancer diagnosis out of concern for being gossiped about in the community. Family members discussed both the patient's and their own concerns regarding open disclosure of BC diagnosis.

> "Maybe there some people who are genuine, but there some people who will go around saying, oh, you know, she's really sick and she has breast cancer. … They think that oh, that's the end of it. I don't like the way people think about it."
>
> *(Family of ND, 45, Female)*

### Psychological domain: "I just saw darkness" (MS, 44, Female)

### Reaction to diagnosis

Upon getting a BC diagnosis, patients, family members and HCPs often recalled and witnessed initial feelings of shock, disbelief and a fear of dying.

> "What came into my mind is just death…. I didn't feel like I was going to live again. I was so much in denial. I was so stressed. When I look back into my family, the extended family, there's never been a diagnosis like that. It was so stressful."
>
> *(ND, 37, Female)*

> "When she called me, I was shocked; I went through depression. I couldn't believe my sister whom I love, so I was shocked. [I had] no appetite, nightmares, sleepless nights."
>
> *(Family of ND, 43, Female)*

MS patients would often describe their situation as traumatic or difficult to accept. This group discussed depression, loss of hope, negative outlooks on life and a greater fear of death.

> "I went back to the doctor again, the breast doctor, and she told me "I'm sorry, you have breast cancer', and it is metastatic and it has spread everywhere. And at that point, I realized this is the end of me."
>
> *(MS, 45, Female)*

In contrast, some ND patients described initially being in shock after receiving the BC diagnosis but reported no major mental health effects. This group of patients typically expressed hope and a readiness to move forward with treatment.

> "I saw death because that is how the media and everyone sees it. […] Your cancer may not be aggressive, but they keep on telling you it might be and go to other parts. So, I think at that point is why I see death but now me being in the situation right now [early diagnosis], I think I see a lot of hope."
>
> *(ND, 26, Female)*

### Financial and work concerns

Direct and indirect treatment-related costs as well as existing living costs, especially in the context of job earnings, and a lack of detailed information from the hospital, acted as a large burden and source of stress for patients and their families. While insurance lessened the burden for some patients, many resorted to family or community support to cover the costs. Regarding workability, some patients either continue to work, take time off, or stop working. Leaving a job was linked to pain or patient discomfort surrounding their appearance after treatment or to focus on treatment.

> "Her greatest worry has been financial. She's covered. But the insurance companies will say they're not paying for this and they're not paying for that. So that is a big worry for her."
>
> *(Family of ND, 45, Female)*

> "Had some patients that go into work every day as normal, and then just come in for their treatments. … Some people maybe they lose their hair and they're not comfortable with how they look so they work from home"
>
> *(HCP, 48, Male)*

### Changes in physical appearance

Hair and weight loss side effects from cancer treatment acted as contributors to poor self-esteem and mental strain in patients.

> "A lot of the women I've spoken to are worried about their appearance. And this is something that they hold quite dear, that they're losing a breast or they're losing hair, they look different from before they started chemotherapy."
>
> *(HCP, 30, Female)*

### Self-worth

Regarding their feelings after BC diagnosis, some patients commented on not having any initial feelings, whereas others described a loss of self-worth or feelings of hope and acceptance. Some MS patients explained that loss of self-worth was induced by poor reactions of others following disclosure and feeling like a burden. Alternatively, some patients expressed a newfound sense of self-worth following diagnosis.

> "The loss of self-worth came because when you tell people you have breast cancer, you realize that in their eyes you become more of a burden. When you call they're not expecting you to say hi they're expecting you to say you're in pain and need help with something."
>
> *(MS, 33, Female)*

*"It's made me more relaxed, and it's made me a better person. It's made me realize the worth of being alive."*

                                                              (MS, 46, Female)

### Spiritual domain: "God put me on this journey … But I don't know why" (MS, 44, Female)

#### Religion as a source of hope

When asked about sources of comfort, all participant groups remarked that patients, regardless of their staging, use religion as a way of staying positive during their treatment. Praying during treatment often provided a sense of peace that comforted patients and their families.

*"We have patients who are very spiritual, they believe they have hope. And they believe that God will heal them."*

                                                              (HCP, 29, Male)

#### Questioning of religion

Conversely, after receiving their diagnosis, some patients started to question their religion.

*"It has made me question my God at some point. Like, 'I'm your daughter, why are you allowing this to happen?'"*

                                                              (ND, 26, Female)

### Discussion

This pilot study was designed to explore the effects of BC diagnosis and treatment experiences on the mental health of BC patients diagnosed and seen at AKUHN. By interviewing patients, their family members and HCPs, we found that the impact of BC on mental health in this setting is complex and influenced by a multitude of factors. The impacts were grouped into four domains of the Quality of Life Model: Physical, Social, Psychological and Spiritual well-being.

### Physical domain

Our findings on the physical health outcomes associated with BC and its effects on mental health resonate with findings in other settings. Studies in South Africa have previously identified that BC treatment can cause traumatizing side effects and emotional suffering among patients (Lambert et al., 2020). These impacts were found among patients in the current study, with all participant groups indicating that chemotherapy and surgery are burdensome experiences that cause pain and physical stress to the patient.

The ability of BC patients to work and complete household duties was hindered by poor physical health, potentially leading to stress and feelings of inadequacy. Loss of job and inability to contribute to household income might also result in loss of self-worth, as well as adverse mental and other health outcomes (Hammarström et al., 2021).

### Social domain

Mental health outcomes were also influenced by aspects of social well-being. Patients' fears of disclosure and reactions to their diagnosis were expressed, with both MS and ND fearing being negatively perceived. Studies have found the process of cancer disclosure to be incredibly difficult, with patients often delaying their disclosure out of guilt or to protect others from the severity of their diagnosis (Hilton et al., 2009). Interestingly, this was also mentioned by HCPs in our sample, as they reported that patients were hesitant to confront the reality of their diagnosis and the possibility of leaving their families behind. HCPs reflected that they do not have the skills to offer adequate counseling to patients following diagnosis. These findings have strong implications for HCPs to be more equipped to react to the emotional needs of patients after diagnosis.

Our interviews indicate that strong family support positively impacts BC patients' well-being. The traditional African family structure can be comprised of joint families or children living with their parents (Abuya et al., 2019). It is common for families to maintain close ties with relatives by visiting or providing socio-economic support (Abuya et al., 2019). This was evident in family member interviews, as family members mentioned contributing to treatment costs, taking patients to hospital visits and having patients live with them at home. Hence, family members are key sources of support for BC patients and the implication exists to better understand their support needs.

All participant groups remarked on spousal abandonment fears due to negative perceptions surrounding BC patients. This finding reflects local gender roles, as in Kenya, a cultural expectation exists for women to bear children and men typically want larger families (Britton et al., 2021; Harrington et al., 2016). Failure to meet these expectations due to BC related barriers serves as a basis for stigmatizing behaviors (Agyemang et al., 2023). The role of gender norms has been relatively underexplored in BC literature in SSA. However, fears of spousal abandonment have been found in SSA communities in Zambia for epilepsy patients or in Uganda for those with HIV, shedding light on social concerns related to illness disclosure (Millum et al., 2019; Nordberg et al., 2020). Although the literature is limited in high-income settings, cancer-related spousal separation was indicated in 57.4% of relationships within a German cohort (Nalbant et al., 2021). Therefore, it seems crucial to understand the influence of cancer on spousal relationships in these varying settings.

### Psychological domain

Psychological distress can be overwhelming following cancer diagnosis, with patients experiencing depression, anxiety and maladaptive coping behaviors (Negussie et al., 2023). This distress was heightened by concern about finances, as the cost of treatment in Kenya is a heavy burden on patients and their family members (Mustapha et al., 2020). The limited scope of National Health Insurance coverage creates an additional burden of undergoing treatment and impacts patients and family members who assist with treatment costs (Mustapha et al., 2020).

Appearance concerns were discussed in all interview groups, with poor body image following chemotherapy and surgery adding to patient stress and loss of womanhood. This is also reflected among cancer patients in Ghana, as changes in appearance contributed to poor self-esteem and quality of life (Iddrisu et al., 2020).

We also observed greater feelings of shock and poorer outlooks on life among MS as compared to ND patients in our study. Likely, the heightened severity of their diagnosis, coupled with social constraints and fears surrounding disclosure contribute to increased psychological impacts. The negative mental health effects identified are concordant with previous studies in Africa. For instance, incidence of cancer-related psychological issues is on

the rise in Ethiopia, where depression and anxiety rates of patients with various cancer types were 58.8% and 60%, respectively (Belay et al., 2022).

### Spiritual domain

There was an increased reliance on religion for some patients, as it was reported that patients would often pray and ask God to heal them. This reliance on religion is typical of those in difficult circumstances, as spirituality can give new meaning to a situation (Leão et al., 2021).

Conversely, some patients had difficulties processing their illness and questioned their religion. It has been reported elsewhere that BC patients can feel abandoned by God and experience related negative emotions (Leão et al., 2021). In a highly religious and spiritual society such as Kenya, the role of religion in BC patients' quality of life should not be under-emphasized (Goodman et al., 2022).

### Limitations

This study has some limitations. Although study participants provided insightful views on their BC diagnosis and care experiences in this Kenyan setting, they represent a relatively well-educated population at a private hospital in urban Kenya. Although AKUHN has a mix of patients in terms of socio-economic backgrounds, patients of low-income backgrounds and from rural areas in Kenya with BC are not significantly represented. Additionally, data from one private institution in Kenya are not sufficient to generalize about BC patients in other parts of the country.

While we divided patients into categories of MS or ND, the severity of cancer symptoms and side effects can vary for each patient, which could impact their physical, social, psychological and spiritual experiences. Additionally, we relied on self-reported experiences of psychological health, which may differ from clinical assessments. This study also utilized quantitative survey methods as part of a larger mixed methods study, which will be analyzed and reported separately.

### Conclusion

The findings from this qualitative study indicate that BC diagnosis has quality of life and mental health effects on BC patients in Nairobi, Kenya. Negative attitudes towards BC in society and burdens of BC care may influence personal relationships and create mental health consequences that are potentially detrimental to the mental health and quality of life of patients and their families. These findings have strong implications for HCPs and family members to be better equipped to provide appropriate support. Policies and services catered towards patients with BC in Nairobi and similar settings, tailored for the specific cultural, social, and healthcare context are encouraged. More research and intervention programs seeking to improve mental health outcomes of patients from varying socioeconomic backgrounds are necessary in Africa.

**Open peer review.** To view the open peer review materials for this article, please visit http://doi.org/10.1017/gmh.2024.79.

**Data availability statement.** The data that support the findings of this study are available from the corresponding author (DA) upon reasonable request.

**Acknowledgements.** We would like to thank the study participants, study staff, the UAB Sparkman Center for Global Health, UAB School of Public Health and the Aga Khan University Hospital in Nairobi, Kenya for contributing to this study.

**Author contribution.** AH, JT, MS, DA, LG and KO contributed to the study design and supported the methodology. KS, IA and NK supported the implementation of the study. DA wrote the first draft of the manuscript. DA, NG and IO conducted study interviews. DA and AH analyzed the collected data. All authors contributed to thematic analysis and commented on manuscript drafts. All authors read and approved the final manuscript.

**Financial support.** This study was supported by a University Research Council (URC) grant from the Aga Khan University, and pilot grants from the Sparkman Center for Global Health and the School of Public Health, University of Alabama at Birmingham, US.

**Competing interest.** The authors declare no competing interests exist.

**Ethics statements.** Ethical approvals were obtained from the Aga Khan University, Kenya, and the University of Alabama at Birmingham, USA. All study participants provided written informed consent. Clear explanations were given to study participants that study participation was voluntary and separate from their employment/medical care.

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
