## [Editor Report]

The paper addresses the mental health aspects of breast cancer in the Kenyan setting. It was conducted at an exclusive hospital that is patronize by the upper end clientele well-resourced private patients. However, the authors brought up this limitation at the end of their discussion. Being a qualitative study, it delves into nuances that could only be resolved by a qualitative study. The paper makes a contribution.

---

## [Editor Report]

The greatest short coming of this paper as far as the Kenyan situation is concerned is the exclusive manner in which the authors do not mention other centres and institutions in Kenya that have been doing and are still doing great jobs on cancer, including breast cancer. The tertiary hospitals include Kenyatta University Teaching and Referral Hospital in Kenya which has a vibrant research centre and the Kenyatta National Hospital Teaching and Referral Hospital which is also the teaching hospital for the Faculty of Medicine of the University of Nairobi. These are two different public hospitals in Kenya which treat a wide spectrum of cancer including breast cancer, not to mention private facilities such as Aga Khan Hospital in Nairobi with very vibrant cancer centre. Indeed, the Kenya Government is very active in the area of cancer treatment and awareness as exemplified by the task force on cancer actually led by a senior psychiatrist. There are other organizations supported by the Government that do advocacy on cancer awareness. I am sure the authors can establish all of these facts by a simple search on what is going on the ground. It is also important for the authors to be explicit on the knowledge gap globally and locally that they are trying to bridge.

The authors have attempted to discuss the limitation of the study, but they could be more explicit. Their center deals with fairly well resourced clientele. This limitation could be improved to bring out the differential response between well resourced clientele and the overwhelming majority of Kenyans who get diagnosed too late because of social-economic factors and therefore the possibility of different responses to the diagnoses and when it is made. It is not possible to extrapolate their findings to the rest of Kenya. This therefore requires to modify the title of their paper to limit it to Aga Khan Hospital and not Kenya in general. 

Secondly, still on limitation, this is a qualitative study with all its shortcoming. However, it could have been useful as a way of explaining nuances in the findings of a quantitative study using well tested instruments such as the WHO Quality of Life to see what best explains the differences between those who are affected by the diagnoses and those who were NOT affected to the same degree by the same diagnoses. This limitation needs to be expanded and highlighted. Mixed method would have been most ideal. 

I would therefore give the authors the benefit of the doubt and recommend major revision short of a reject.

---

## [Editor Report]

The authors have adequately addressed the issues I raised and in particular the limitations of the study.